# Neural basis of somatosensory target detection independent of uncertainty, relevance, and reports

Pia Schröder*, Timo Torsten Schmidt, Felix Blankenburg

Neurocomputation and Neuroimaging Unit, Freie Universität Berlin, Berlin, Germany

**Abstract** Research on somatosensory awareness has yielded highly diverse findings with putative neural correlates ranging from activity within somatosensory cortex to activation of widely distributed frontoparietal networks. Divergent results from previous studies may reside in cognitive processes that often coincide with stimulus awareness in experimental settings. To scrutinise the specific relevance of regions implied in the target detection network, we used functional magnetic resonance imaging (n = 27) on a novel somatosensory detection task that explicitly controls for stimulus uncertainty, behavioural relevance, overt reports, and motor responses. Using Bayesian Model Selection, we show that responses reflecting target detection are restricted to secondary somatosensory cortex, whereas activity in insular, cingulate, and motor regions is best explained in terms of stimulus uncertainty and overt reports. Our results emphasise the role of sensory-specific cortex for the emergence of perceptual awareness and dissect the contribution of the frontoparietal network to classical detection tasks.

DOI: https://doi.org/10.7554/eLife.43410.001

*For correspondence:
pia.schroeder@fu-berlin.de

Competing interests: The authors declare that no competing interests exist.

## Introduction

The target detection task is a standard paradigm to study the neural correlates of perceptual awareness. In the near-threshold detection task, participants are presented with stimuli at intensities close to their individual detection thresholds, resulting in detection rates of ~50%. Neural responses underlying detected and undetected targets can then be contrasted at identical physical stimulation parameters to identify the neural correlates of conscious access.

In the somatosensory domain, research on this task has identified a range of areas that correlate with target detection. These include the thalamus, primary (SI) and secondary (SII) somatosensory cortices, motor areas, the anterior insular cortex (AIC), anterior cingulate cortex (ACC), as well as posterior parietal and prefrontal regions (*Auksztulewicz et al., 2012*; *Bastuji et al., 2016*; *Allen et al., 2016*; *Bornhövd et al., 2002*; *Büchel et al., 2002*; *de Lafuente and Romo, 2005*; *de Lafuente and Romo, 2006*; *Frey et al., 2016*; *Hirvonen and Palva, 2016*; *Jones et al., 2007*; *Moore et al., 2013*). This diversity of findings parallels results from the visual and auditory modalities (e.g. *Carmel et al., 2006*; *Eriksson et al., 2007*; *Rees et al., 2002*) and underlies the idea that perceptual awareness emerges when local information is propagated from sensory cortices to higher order brain regions to elicit a reverberating network of broadcast activity (*Baars, 1997*; *Dehaene et al., 2006*).

While a complex network activation may seem a suitable candidate to explain a complex phenomenon such as perceptual awareness, the functional specificity of the identified regions remains largely unknown. One problem that complicates the interpretation is that classical target detection tasks not only probe perceptual awareness but involve a range of correlated cognitive processes that may confound classical contrastive analyses (*Aru et al., 2012*; *de Graaf et al., 2012*). In the

context of the somatosensory near-threshold detection task, four aspects are particularly problematic: 1. perception of near-threshold stimuli is difficult, and resolution of associated uncertainty and introspective processes may differ between detected and undetected targets (*de Lafuente and Romo, 2011*). 2. Target detection is the explicit behavioural goal of the task and therefore, detected targets have higher behavioural relevance than undetected targets (*Farooqui and Manly, 2018*). 3. Target detection is directly mapped to overt reports that allow for assessment of participants' trial-by-trial perception (*Tsuchiya et al., 2015*). 4. Overt reports are often communicated with button presses by one hand while stimulation occurs on the other hand, which may affect cortical excitability in homologue regions of the sensorimotor homunculus (*Zagha et al., 2013*). All these variables potentially contribute to the commonly observed network activation and dedicated experimental paradigms are warranted to scrutinise its functional specificity for target detection and accordingly, somatosensory awareness.

Here, we used functional magnetic resonance imaging (fMRI) on a novel somatosensory detection task that explicitly varies stimulus uncertainty and controls for behavioural relevance, overt reports, and motor responses. To distinguish BOLD responses reflecting target detection from those indicating concomitant processes, we fit simple behavioural models to our fMRI data, capturing trial-wise physical stimulus intensity, target detection, detection probability, expected uncertainty, and overt reports, respectively. We dissociate corresponding representations in the brain by means of Bayesian Model Selection (BMS, Stephan, Penny, Daunizeau, Moran, & Friston, 2009), which determines which model best explains the data in every voxel of the brain based on model evidence maps. Building on insights from the visual modality (*Farooqui and Manly, 2018*; *Frässle et al., 2014*; *Koch et al., 2016*), we hypothesise that BOLD responses associated with target detection are restricted to somatosensory regions, whereas activity in the frontoparietal network reflects cognitive processes that follow from task requirements.

## Results

### Experimental paradigm

Participants performed a two-alternative forced choice somatosensory detection task on electrical pulses administered to their left median nerves inside the fMRI scanner (see *Figure 1* and Materials and methods for a detailed description of the task design). To vary stimulus uncertainty, we presented stimuli at ten different intensity levels that were individually adjusted to sample the full dynamic range of each participant's psychometric function from 0 to 100% detectability. Accordingly, stimuli presented near individual 50% detection thresholds were expected to be associated with higher uncertainty (as defined by larger trial-by-trial variability in target detection) than clearly sub- or supraliminal stimuli. To balance behavioural relevance and overt reports across detected and undetected targets, instead of directly reporting target detection, participants were required to match their perception of somatosensory target stimuli against simultaneously presented visual cues that signalled stimulus presence or absence. As a result, detected and undetected targets could result in the same overt report (match or mismatch), making them equally relevant for the task. Finally, to avoid motor response-related activation of cortical hand representations, instead of giving manual responses, participants responded with saccades to peripheral response cues. We analysed 3T fMRI data of n = 27 participants who performed 4 runs of 100 experimental trials each.

### Behaviour

Participants detected 52.25 ± 11.65% (mean ± standard deviation) of targets. Note that targets were presented on every trial and only the stimulus intensity was individually manipulated to render targets sub- or supraliminal. Therefore, the detection rate was identical to the hit rate and false alarms or correct rejections could not occur. As expected, participants' target detection varied with stimulus intensity, resulting in characteristic sigmoidal psychometric functions (*Figure 2*). A Bayesian equivalence of the paired-sample t-test suggests strong evidence for shorter reaction times for detected than undetected targets (detected: 352.11 ± 50.98 ms, undetected: 363.88 ± 55.97 ms, difference: 11.77 ± 14.84 ms, BF10 = 88.01). To test if the task manipulation successfully dissociated target detection from overt reports, we computed Bayesian tests of association for all participants and found that none of the participants showed positive evidence for an association between the two

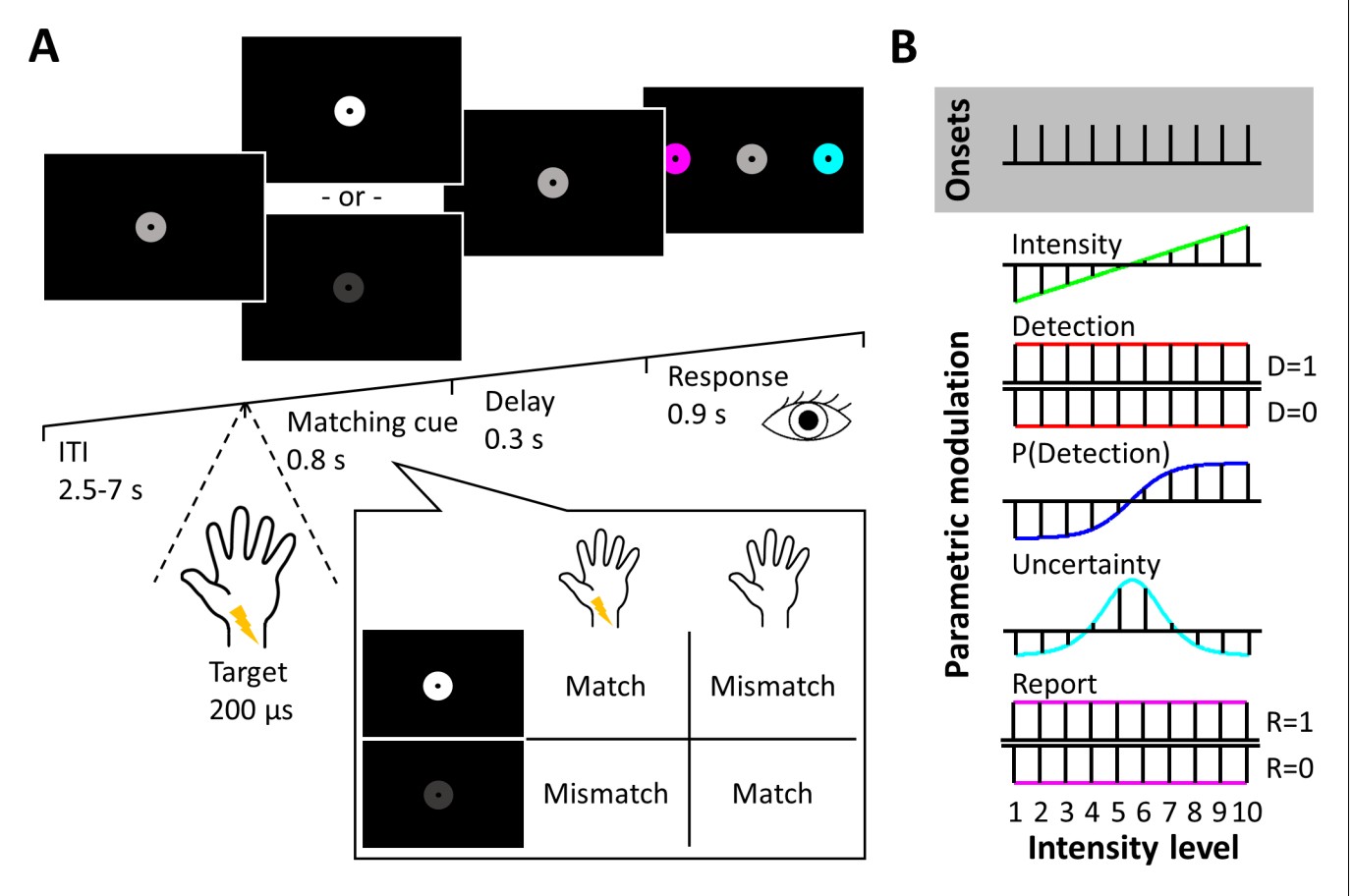

**Figure 1.** Experimental design. (**A**) Trial design. After a variable intertrial interval of 2.5–7 s, electrical target stimuli and visual matching cues were presented simultaneously. A white matching cue signalled stimulus presence, a dark grey matching cue signalled stimulus absence. After a short delay, participants reported a match or mismatch between the cue and their somatosensory percept by selecting one of two colour-coded disks with a saccadic eye movement. Example: If participants detected the target and saw a white matching cue, they would report a match. Likewise, if they did not detect the target and saw a dark grey matching cue, they would also report a match, resulting in the same behavioural relevance of detected and undetected targets and orthogonalisation of target detection and overt reports. (**B**) Graphical depiction of experimental regressors plotted against stimulus intensity levels. Five stimulus and behavioural dimensions of our task were specified as parametric regressors on trial onsets: physical stimulus intensity, target detection, detection probability, expected uncertainty, and overt reports.

DOI: https://doi.org/10.7554/eLife.43410.002

variables (all BF10 <.5). In contrast, we found positive evidence for independence for all participants (BF01 >4) except one (BF01 = 2.62).

## fMRI

From our paradigm, we extracted five experimental regressors to identify brain regions that process the various stimulus and behavioural dimensions of the task: linear stimulus intensity, binary target detection, detection probability, expected uncertainty, and overt reports (*Figure 1B*). The shared variance in some of these regressors (intensity, detection, detection probability) would result in multicollinearity issues and unstable beta estimates in classical regression analyses. To overcome this problem, we separated the regressors into different models and performed Bayesian model comparison to determine which regressor best explained the data over and above shared variance. To this end, the experimental regressors were incorporated in five different general linear models (GLMs) as parametric regressors on trial onsets, each of which constituted a specific hypothesis of expected BOLD responses. We then estimated the models using the first-level Bayesian GLM estimation as implemented in SPM12 (*Penny et al., 2007*). The resulting model evidence maps were subjected to random effects BMS to obtain one exceedance probability (EP) map per model, indicating the voxel-

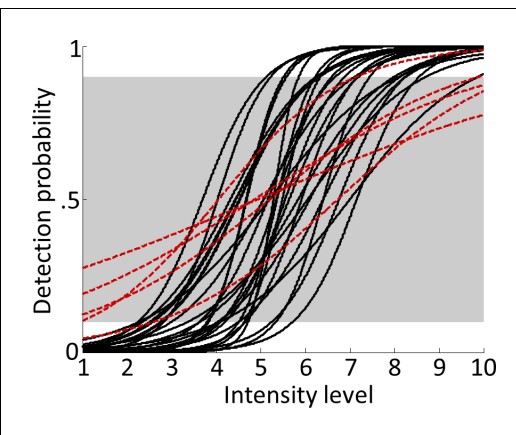

**Figure 2.** Psychometric functions. Logistic functions were fitted to each participant's behavioural data during the main experiment and averaged across runs to obtain continuous models of individual psychometric functions. Note that although the shape of the psychometric function can vary considerably across participants, due to the individually adjusted stimulus intensities, the resulting curves were normalised to span 0–100% detection probability from intensity levels 1 to 10. Red dashed lines show fitted psychometric functions of five participants that failed to reach ≤10% detection probability for intensity level 1 or ≥90% detection probability for intensity level 10 (outside the grey-shaded area) and were therefore excluded from all further analyses.

DOI: https://doi.org/10.7554/eLife.43410.003

The following source data is available for figure 2:

**Source data 1.** Target detection rates for all intensity levels.

DOI: https://doi.org/10.7554/eLife.43410.004

wise probability that a particular model explained the data better than any of the other models (*Rosa et al., 2010*; *Stephan et al., 2009*). To extract only those voxels where one model clearly outperformed all other models, we thresholded these maps at EP ≥ .99 and further inspected only those voxels that exceeded this threshold. When models are expected to share a lot of probability mass due to correlated regressors, they may be too similar in the variance they explain to outperform each other sufficiently to exceed a desired EP threshold. To avoid such model dilution (*Hoeting et al., 1999*), in a first step, we combined the intensity, detection, and detection probability models into a model family (+family) and performed BMS on the family level (*Penny et al., 2010*) to define regions of interest (ROIs), in which the +family yielded EP ≥ .99. The +family models were then assessed individually within these ROIs. The uncertainty and report models on the other hand were not expected to share variance with any of the other models and were therefore assessed in a whole-brain analysis.

The +family defined ROIs in contralateral right SI extending into right superior parietal lobule (SPL), bilateral SII extending into posterior insular cortices, as well as left superior frontal gyrus (SFG), left inferior parietal lobule (IPL), and left visual area V3 (*Figure 3A*). Within these regions, the intensity model was the best model in the anterior part of the right SI cluster (peak voxel [38, −40, 66], peak EP = .96), primarily spanning areas 3b, 1, and 2. Further intensity representations were found in regions of bilateral SII, in particular the most anterior and posterior edges of the right SII cluster (anterior: peak voxel [54, −6, 4], peak EP = .90; posterior: peak voxel [62, −34, 22], peak EP = .98) and the medial part of left SII (peak voxel [−46, –34, 22], peak EP = .90). The detection probability model best explained data in the posterior part of right SI (primarily area 2) extending into SPL (peak voxel [34, −50, 62], peak EP = .99), as well as large regions of bilateral SII (right: peak voxel [56, −16, 20], peak EP = .99; left: peak voxel [−60, –36, 20], peak EP = .98). Finally, the binary detection model was the best model in superior and inferior parts of the right SII cluster (superior: peak voxel [62, −20, 30], peak EP = .95; inferior: peak voxel [52, −22, 8], peak EP = .96) and in mostly lateral regions of left SII (peak voxel [−62, –36, 26], peak EP = .93). Further detection-sensitive regions were found in left SFG (peak voxel [−26, 56, 22], peak EP = 1), left IPL (peak voxel [−50, –58, 46], peak EP = 1), and left V3 (peak voxel [−12, –80, −16], peak EP = 1). Within SI, the spatial distribution of voxels explained by the intensity and detection probability models, respectively was found to follow the known cytoarchitectonic subdivisions of the anterior parietal cortex (*Figure 3—figure supplement 1*). In SII on the other hand, voxels with sensitivity to intensity, detection probability, and detection did not show an apparent organisation along the cytoarchitectonic subdivisions of the parietal operculum.

Having established regions that were well explained by the +family models, we further examined the underlying model parameters. To this end, we extracted beta estimates of the respective experimental regressors from individual BMS peaks and computed Bayes factors quantifying the evidence that these estimates systematically deviated from zero on the group level. Since the model selection procedure does not account for directionality of the underlying effects (i.e. both positive and negative parameter estimates may contribute to the model evidence), we imposed this second constraint

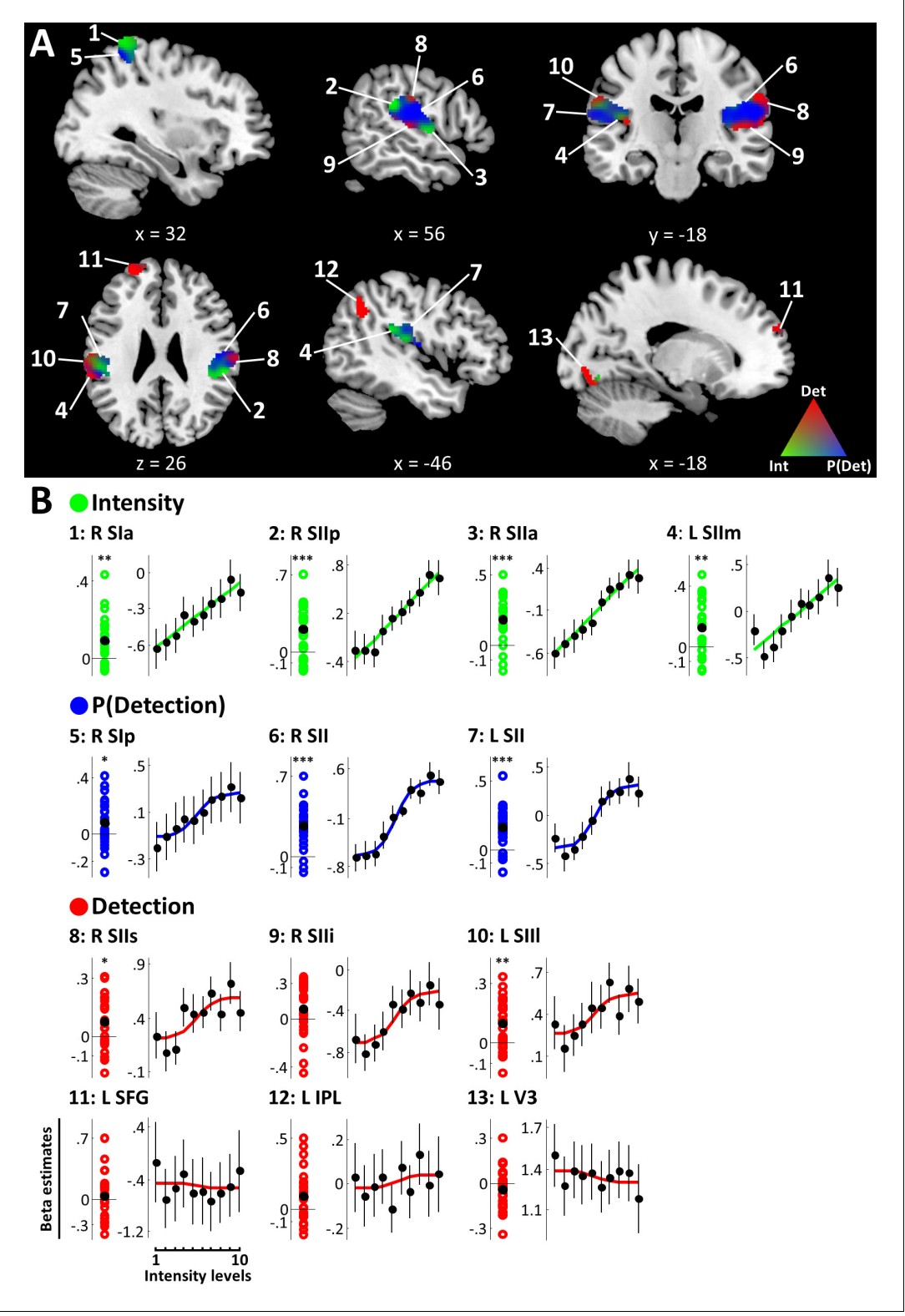

**Figure 3.** +family models. (**A**) BMS results (ROI analysis). EPs of the +family models are displayed within +family ROIs. RGB values indicate model EPs: The corners of the RGB triangle correspond to EP = 1 signifying a clear winner of the BMS, whereas intermixed colours indicate similar EPs for respective models. Intensity (green), P(Detection) (blue), Detection (red). k ≥ 50 voxels. (**B**) Beta estimates and stimulus response profiles. Left panels: Beta estimates of the winning models' experimental regressors were extracted from individual BMS peak voxels. *Figure 3 continued on next page*

*Figure 3 continued*

Each coloured circle corresponds to one participant's beta estimate. Black circles mark group means. Asterisks indicate evidence for a deviation from zero: *BF >3, **BF >20, ***BF >150. Right panels: beta estimates for different intensity levels were extracted from regions of interest and plotted to provide SRPs. For visualisation, fitted representations of the winning models are plotted along with the beta estimates. Error bars represent the standard error of the mean. Somatosensory regions show representations of stimulus intensity, detection probability, and binary target detection, which are reflected in their SRPs. Detection-sensitive regions in prefrontal, posterior parietal, and visual areas do not show systematic relationships with stimulus intensity.

DOI: https://doi.org/10.7554/eLife.43410.005

The following source data and figure supplement are available for figure 3:

**Source data 1.** Beta estimates for clusters defined by the +family models.

DOI: https://doi.org/10.7554/eLife.43410.007

**Figure supplement 1.** Distribution of models in cytoarchitectonic subregions of SI and SII.

DOI: https://doi.org/10.7554/eLife.43410.006

---

on the identified regions to detect systematic variation across participants and thus, systematic association of the behavioural regressors with the observed BOLD activity. To visualise these associations, we extracted stimulus response profiles (SRPs). SRPs show the fitted activity of a region as a function of stimulus intensity and can provide a visual representation of the model fit. Regions that are well explained by models featuring a systematic relationship with stimulus intensity (intensity, detection, detection probability, uncertainty) were expected to show a response profile visibly reflecting this relationship. Regions that are explained by models with no such relationship on the other hand (report) were expected to show no systematic differences in BOLD responses across stimulus intensities.

All regions explained by the intensity model showed strong (R SIa: $BF_{10}$ = 33.94; L SIIm: $BF_{10}$ = 81.58) or very strong (R SIIp: $BF_{10}$ = 385.13; R SIIa: $BF_{10}$ = 6114.09) evidence for a positive regression weight of the intensity regressor indicating a positive relationship of stimulus intensity and BOLD responses in these regions (*Figure 3B*, *Table 1*). This result is reflected in the SRPs of intensity-sensitive regions, which show a linear response increase with increasing stimulus intensity (*Figure 3C*). Likewise, positive regression weights for the detection probability regressor in SI and SII were confirmed with positive (R SIp: $BF_{10}$ = 3.04) and very strong evidence (R SII: BF = 232795.89, L SII: BF = 7961.84) and, correspondingly, these regions show sigmoidal SRPs. For the binary detection regressor, positive regression weights were exclusively found in bilateral SII (R SIIs: BF = 8.63; L SIII: BF = 96.7). Parameter estimates in all other regions explained by the detection model failed to yield positive evidence for a systematic deviation from zero (R SIIi: BF = 1.36; L SFG: BF = .31; L IPL: BF = 2.84; L V3: BF = .65), indicating that in these regions the response to target detection was not systematic across participants. Accordingly, the sigmoidal shape of the SRP that would be expected from an area showing a categorical response to target detection (recall that target detection averaged over trials and plotted against stimulus intensity amounts to the psychometric function) is less pronounced in these regions.

The uncertainty model yielded EP $\geq$ .99 in bilateral superior medial gyrus (SMG) extending into ACC (peak voxel [2, 30, 40], peak EP = 1) and bilateral AIC (right: peak voxel [36, 22, -6], peak EP = 1; left: peak voxel [−34, 18, −8], peak EP = 1) (*Figure 4A*, *Table 1*). Very strong evidence confirms positive beta estimates in all these regions (SMG/ACC: $BF_{10}$ = 366797.07; R AIC: $BF_{10}$ = 453.09; L AIC: $BF_{10}$ = 479302.70), and the respective SRPs show a clear inverse U-shaped function, suggesting that these areas exhibit the strongest activity for targets close to detection thresholds, when expected uncertainty is highest (*Figure 4B*).

Finally, the report model fit best in left supplementary motor area (SMA) (peak voxel [−2, 8, 64], peak EP = 1), right supramarginal gyrus (SMarG) (peak voxel [60, -34, 44], peak EP = 1), and left thalamus (peak voxel [−6, −16, 10], peak EP = 1) (*Figure 5A*, *Table 1*). Of these, SMA was found to be the only region showing beta estimates that reliably deviated from zero (SMA: $BF_{10}$ = 65.14). Beta estimates extracted from left thalamus and right SMarG did not yield positive evidence for an effect (L Thal: $BF_{10}$ = .22; R SMarG: $BF_{10}$ = .23) and these regions' sensitivity to overt reports is therefore considered unsystematic. As expected from the lack of association between overt reports and target

**Table 1.** Brain regions showing EP $\geq$ .99 for any of the tested models.

For the +family models the .99 EP threshold was applied on the family level and individual peak EPs are reported for every model. k $\geq$50 voxels. Betas of experimental regressors extracted from individual BMS peaks are reported as mean ± SEM. ACC: anterior cingulate cortex, AIC: anterior insular cortex, IPL: inferior parietal lobule, SI: primary somatosensory cortex, SII: secondary somatosensory cortex, SFG: superior frontal gyrus, SMA: supplementary motor area, SMarG: supramarginal gyrus, SMG: superior medial gyrus. a: anterior, p: posterior, i: inferior, s: superior, m: medial, l: lateral.

| Cluster size | Region | Peak MNI (x,y,z) | | | Peak EP | Beta | BF10 |
|---|---|---|---|---|---|---|---|
| *Intensity* | | | | | | | |
| 247 | R SIa (BA 3b, 1, 2) | 38 | −40 | 66 | .96 | .09 ± .02 | 33.94 |
| 276 | R SIIp | 62 | −34 | 22 | .98 | .18 ± .03 | 385.13 |
| 213 | R SIIa | 54 | −6 | 4 | .90 | .20 ± .04 | 6114.09 |
| 212 | L SIIm | −46 | −34 | 22 | .90 | .13 ± .03 | 81.58 |
| *Detection probability* | | | | | | | |
| 71 | R SIp (BA 2) | 34 | −50 | 62 | .99 | .08 ± .03 | 3.04 |
| 932 | R SII | 56 | −16 | 20 | .99 | .27 ± .04 | 232795.89 |
| 602 | L SII | −60 | −36 | 20 | .98 | .17 ± .03 | 7961.84 |
| *Detection* | | | | | | | |
| 189 | R SIIi | 52 | −22 | 8 | .96 | .09 ± .04 | 1.36 |
| 76 | R SIIs | 62 | −20 | 30 | .95 | .08 ± .03 | 8.63 |
| 128 | L SIII | −62 | −36 | 26 | .93 | .10 ± .02 | 96.70 |
| 116 | L SFG | −26 | 56 | 22 | 1 | .04 ± .05 | .31 |
| 66 | L IPL | −50 | −58 | 46 | 1 | .09 ± .03 | 2.84 |
| 72 | L V3 | −12 | −80 | −16 | 1 | −.04 ± .02 | .65 |
| *Uncertainty* | | | | | | | |
| 664 | SMG/ACC | 2 | 30 | 40 | 1 | .33 ± .04 | 366797.07 |
| 127 | R AIC | 36 | 22 | −6 | 1 | .22 ± .03 | 453.09 |
| 70 | L AIC | −34 | 18 | −8 | 1 | .22 ± .05 | 479302.70 |
| *Report* | | | | | | | |
| 132 | L SMA | −2 | 8 | 64 | 1 | −.12 ± .03 | 65.14 |
| 71 | L Thalamus | −6 | −16 | 10 | 1 | −.01 ± .02 | .22 |
| 51 | R SMarG | 60 | −34 | 44 | 1 | .02 ± .04 | .23 |

DOI: https://doi.org/10.7554/eLife.43410.008

detection, SRPs in these areas did not reflect a systematic relationship with stimulus intensity (*Figure 5B*).

## Discussion

To scrutinise the neural processes underlying somatosensory target detection in humans, we employed an experimental paradigm that explicitly dissociates target detection from stimulus uncertainty, behavioural relevance, overt reports, and motor responses. Using Bayesian Model Selection on the acquired fMRI data, we observe a transformation from physical to perceptual representations as the target is propagated through the somatosensory hierarchy. This transformation primarily occurred in SI and SII, whereas expected uncertainty was represented in insular and cingulate regions and overt reports were processed in supplementary motor cortex. Our analysis reveals large overlap with the previously identified target detection network but assigns functional specificity to the involved regions.

SI is the first cortical region to receive somatosensory input from the contralateral body side. It is subdivided into four somatotopic maps, areas 3a/b, 1, and 2, that are organised along an anterior-

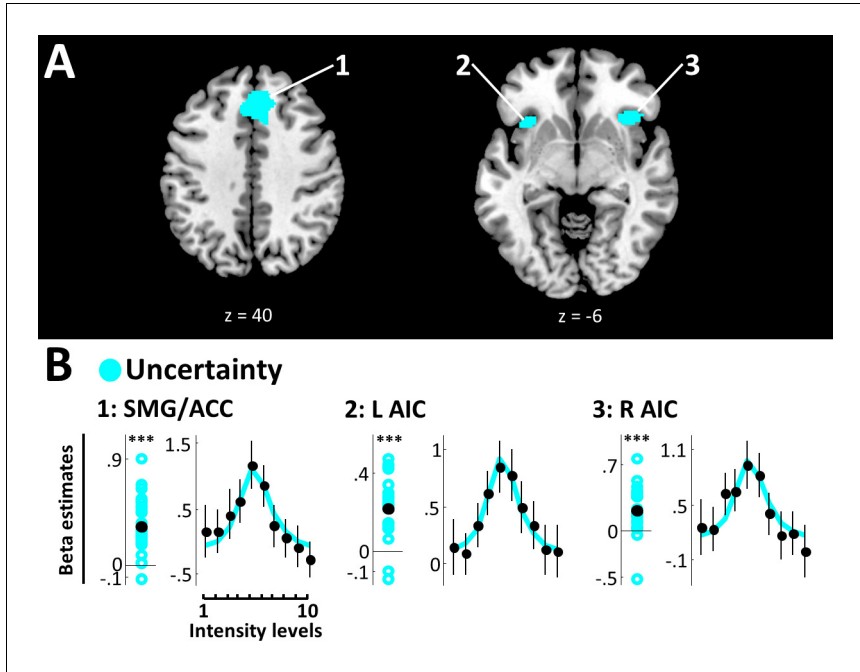

**Figure 4.** Uncertainty model. (**A**) BMS results (whole-brain analysis). Voxels with EP ≥.99 for the uncertainty model are displayed. Expected uncertainty best modelled data in bilateral SMG/ACC and bilateral AIC. k ≥50 voxels. (**B**) Beta estimates and stimulus response profiles. Beta estimates of the winning models' experimental regressors (left panels) and SRPs (right panels) are displayed as in *Figure 3*. *BF >3, **BF >20, ***BF >150. SMG/ACC and AIC show positive beta estimates and clear inverse U-shaped SRPs, confirming a representation of stimulus uncertainty.
DOI: https://doi.org/10.7554/eLife.43410.009

The following source data is available for figure 4:

**Source data 1.** Beta estimates for clusters defined by the uncertainty model.
DOI: https://doi.org/10.7554/eLife.43410.010

posterior axis on the postcentral gyrus and increase in their hierarchical level (*Delhaye et al., 2018*; *Felleman and Van Essen, 1991*). Our analysis suggests that during somatosensory target detection, contralateral areas 3b, 1, and anterior parts of area 2 process the physical stimulus intensity, whereas more posterior parts of area 2, extending into adjacent SPL, represent the probability to detect a target. Previous research has been ambiguous regarding the role of SI for target detection. Some studies found no difference in SI responses for detected and undetected targets (*de Lafuente and Romo, 2005*; *Schubert et al., 2006*; *Wühle et al., 2010*; *Wühle et al., 2011*) and are in line with a representation of physical stimulus properties. Others have reported stronger SI evoked magneto-/ electroencephalography responses for detected compared to missed stimuli (*Hirvonen and Palva, 2016*; *Jones et al., 2007*; *Palva et al., 2005*). Interestingly, most studies reporting early detection-related effects in SI used the classical near-threshold detection task, whereas those that did not find effects in SI typically used alternative approaches (varying stimulus intensities: *de Lafuente and Romo, 2005*; *de Lafuente and Romo, 2006*; backward masking: *Schubert et al., 2006*; paired-pulse paradigms: *Wühle et al., 2010*, *Wühle et al., 2011*). As target detection at perceptual thresh-old changes with fluctuations in cortical excitability (*Boly et al., 2007*; *Frey et al., 2016*; *Moore et al., 2013*; *Schubert et al., 2009*; *Weisz et al., 2014*), this dissociation raises the question, whether differentiable SI responses for detected and undetected near-threshold stimuli are in fact markers of stimulus awareness or rather the result of background processes that may facilitate or attenuate target detection depending on pre-stimulus brain states (*Schubert et al., 2006*). In our data, we do observe a transformation of stimulus representations in SI as the stimulus is propagated up the local hierarchy. However, although the detection probability model captures some perceptual properties of the stimulus and may constitute a first step towards perceptual readout, it does not explicitly differentiate between detected and undetected trials and importantly, predicts the same

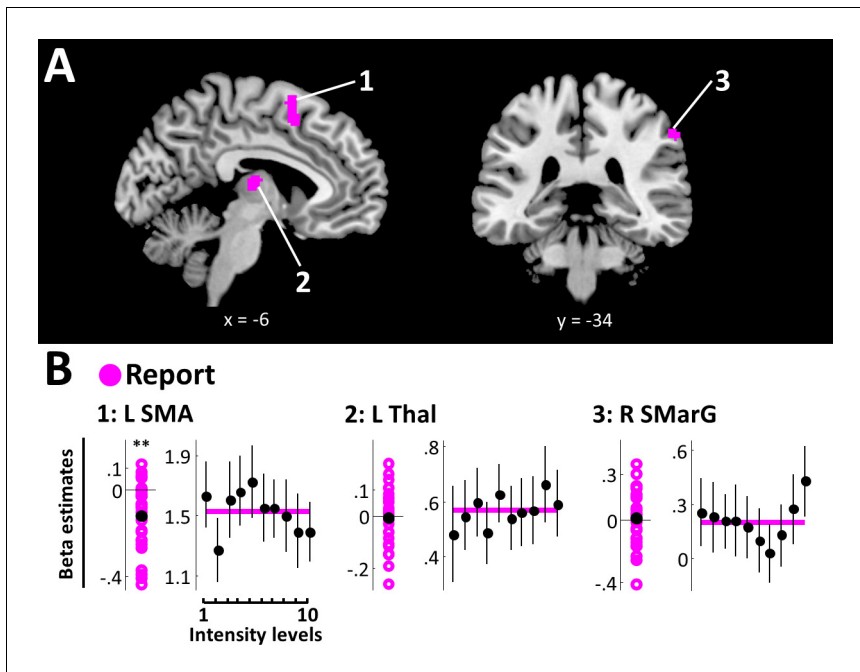

**Figure 5.** Report model. (**A**) BMS results (whole-brain analysis). Voxels with EP ≥.99 for the report model are displayed. Overt reports best modelled data in left SMA, left thalamus, and right SMarG. k ≥50 voxels. (**B**) Beta estimates and stimulus response profiles. Beta estimates of the winning models' experimental regressors (left panels) and SRPs (right panels) are displayed as in *Figure 3*. *BF >3, **BF >20, ***BF >150. L SMA is the only report region that shows beta estimates that systematically deviate from zero. None of the identified report regions show systematic relationships with stimulus intensity, as expected from the lack of association between overt reports and target detection.

DOI: https://doi.org/10.7554/eLife.43410.011

The following source data is available for figure 5:

**Source data 1.** Beta estimates for clusters defined by the report model.

DOI: https://doi.org/10.7554/eLife.43410.012

level of activation for threshold stimuli, regardless of whether they were perceived or not. Therefore, our results do not support detection-sensitivity in SI.

Responses in bilateral SII mirror the transformation from stimulus intensity to detection probability in SI and importantly, they also show an effect of binary target detection in adjacent regions. The distribution of these effects did not align with the cytoarchitectonic subdivisions of the parietal operculum, whose distinct functional roles remain largely elusive. The less pronounced somatotopy and large interindividual variability in SII (*Eickhoff et al., 2006a*; *Sanchez Panchuelo et al., 2018*) may have concealed the exact organisation of the observed effects and further studies are needed to ascertain the specific functional topology. Nonetheless, the relevance of region SII for somatosensory target detection is well documented in the literature. It exhibits stronger somatosensory evoked potentials (*Auksztulewicz et al., 2012*; *Wühle et al., 2010*; *Wühle et al., 2011*), increased spike rates (*de Lafuente and Romo, 2006*), stronger BOLD-signal changes (*Moore et al., 2013*), and increased network integration (*Weisz et al., 2014*) when stimuli are detected compared to when they are missed. Our findings are in line with intracortical recordings in monkeys (*de Lafuente and Romo, 2006*) showing that SII neurons are the first to be predictive of perceptual decisions. Previous studies have further shown that SII integrates information from different body parts (*Goldin et al., 2018*), indexes unexpected somatosensory stimuli (*Chen et al., 2008*) and stimulus omissions (*Andersen and Lundqvist, 2019*), and can adapt to task requirements (*Romo et al., 2002*). In combination with the observed shift from physical to perceptual representations in our study, evidence suggests that SII might be a central relay point at which behaviourally relevant stimuli are transformed into perceptual outcomes.

Expected uncertainty, here defined as susceptibility to perceptual fluctuations, best explained the data in SMG/ACC and AIC. These regions are commonly referred to as the salience network, which is involved in conflict monitoring, reorienting of attention, and introspective processes (*Menon and Uddin, 2010*) and, interestingly, was found to be sensitive to perceptual ambiguity (*Lamichhane et al., 2016*). Importantly, both ACC and insular cortex have previously been implied in somatosensory target detection (*Bastuji et al., 2016*; *Boly et al., 2007*; *Büchel et al., 2002*; *Hirvonen and Palva, 2016*; *Moore et al., 2013*). The insula in particular has been assigned a pivotal role in interoceptive inference (*Seth, 2013*) and awareness (*Craig, 2009*; *Critchley et al., 2004*), although recent research suggests that activity in AIC might reflect interoceptive sensitivity and learning rather than subjective experience per se (*Canales-Johnson et al., 2015*). In light of our results and given the fact that interoceptive signals, such as the heartbeat, are often faint signals, future research may benefit from combining measures of perceptual uncertainty (e.g. *Garfinkel et al., 2015*) with neuroimaging techniques to reconcile interoceptive processing with insular response properties. With regard to its role in somatosensory perception, the AIC has further been demonstrated to show strengthened backward connectivity with somatosensory cortex upon mismatch detection (*Allen et al., 2016*). Considering that in our study AIC particularly activated for hard-to-detect stimuli, these findings suggest that – although it does not generally reflect target detection – AIC may exert top-down modulation on somatosensory cortex to facilitate somatosensory processing.

The primary objective of the target detection task is to uncover neural processes underlying the emergence of perceptual awareness. Our results identify responses in somatosensory cortex as the best correlates of perceptual awareness and are in line with previous research emphasising the role of dedicated sensory regions for both visual (*Boehler et al., 2008*; *Hurme et al., 2017*; *Pascual-Leone and Walsh, 2001*; *Ress et al., 2000*) and somatosensory awareness (*Auksztulewicz et al., 2012*; *Jones et al., 2007*). However, this view is not unopposed and especially the role of prefrontal cortex (PFC) remains a topic of intense debate. While opponents argue that activity in PFC is an artefact of report requirements and behavioural relevance (*Boly et al., 2017*; *Brascamp et al., 2015*; *Farooqui and Manly, 2018*; *Frässle et al., 2014*; *Koch et al., 2016*; *Pitts et al., 2014*; *Tsuchiya et al., 2015*), proponents uphold that PFC has been demonstrated to code perceptual content even in passive paradigms (*Panagiotaropoulos et al., 2012*) and that null-findings in PFC may be the result of insufficient sensitivity in the methods (*Odegaard et al., 2017*). Ours is the first study to explicitly dissociate stimulus awareness from overt reports and behavioural relevance in the somatosensory domain and while we can only draw conclusions to the extent of the stimuli and methods used in this study, our findings do not support reliable representations of somatosensory awareness in PFC.

Importantly, we did not find strictly detection-related responses in posterior parietal cortex either, a result that conflicts with the 'posterior hot zone' theory of conscious perception (*Koch et al., 2016*) but is in line with an earlier finding suggesting that posterior parietal activation is related to goal completion but not stimulus awareness (*Farooqui and Manly, 2018*). Given that we controlled for goal completion effects in our study by equating behavioural relevance across detected and undetected targets, we may conclude that the correlates of target detection are largely restricted to dedicated sensory processing sites when controlled for common task requirements.

The aim of the current study was to experimentally dissociate target detection from four potentially confounding processes and hence, preclude these processes as root causes of detection-related neural activity. 1. Stimulus uncertainty: By varying stimulus intensities across trials, we implicitly tested for specific response profiles that occur in regions reflecting target detection. Specifically, these regions were expected to show low activity for low, subthreshold stimulus intensities and high activity for high, suprathreshold stimulus intensities. Since uncertainty, no matter if subjective or objective uncertainty measures are used, decreases for high-intensity stimuli, we can exclude uncertainty related processes as the driving force of activity in regions showing such a response profile. 2. Behavioural relevance: In order to perform the task, participants had to report a match or mismatch between their target perception and a visual matching cue. Since detected and undetected targets were found to occur equally often for match and mismatch reports, the two perceptual outcomes were equally relevant for the task and thus, differences in behavioural relevance are unlikely to have caused the observed detection-related activity. 3. Overt reports: As mentioned in point 2, overt

reports were independent of target detection and are therefore unlikely to have caused detection-related effects. However, participants might still form a covert but explicit decision regarding the presence or absence of the target prior to making the match/mismatch decision, which could potentially confound the observed detection effects. Two reasons speak against this interpretation: First, the task imposed a time constraint that required participants to form their decisions fast. Therefore, the task could best be solved by directly comparing the two input modalities without engaging intermediate steps, a strategy, that was verbally confirmed by a majority of participants after the experiment. Second, the only areas that showed reliable and systematic covariation with target detection were regions in bilateral SII. An explicit internal decision regarding the absence or presence of a target stimulus would be expected to involve supramodal processes (because the same decision can be made on stimuli from different modalities) and therefore recruit higher order association cortices. Since this was not the case, we consider it unlikely that the detection effects observed in SII were induced by explicit perceptual decisions. Nonetheless, rapid low-level perceptual decision processes cannot be completely excluded and alternative approaches, such as no-report paradigms (*Tsuchiya et al., 2015*), may aid in disentangling their impact. 4. Motor responses: Participants gave their reports by making saccades to peripheral response cues. None of the observed detection effects overlap with regions commonly observed during saccadic eye movements (*Ettinger et al., 2008*; *Kimmig et al., 2001*) and effects in SI were localised to hand areas and far removed from eye representations in the somatosensory homunculus. Changes in cortical excitability due to motor responses are therefore unlikely to have affected the results. In conclusion, we consider the detection-related responses observed in bilateral SII to be largely free of the addressed experimental confounds.

A theoretical framework that accommodates awareness-related responses in sensory cortices is the recurrent processing hypothesis of conscious perception, which argues that perceptual awareness emerges when feedforward signals from early sensory cortex are consolidated by re-entrant feedback (*Lamme, 2006*). Such a mechanism might account for the early transformation from stimulus intensity to detection probability that we observed in SI and in fact, previous research on somatosensory target detection suggests that stimulus awareness is best captured in recurrent interactions between SI and SII (*Auksztulewicz et al., 2012*; *Cauller and Kulics, 1991*; *Kwon et al., 2016*; *Yang et al., 2016*). However, these earlier studies have used simpler versions of the detection task and studies employing methods with high temporal resolution in combination with rigorous experimental control are warranted to scrutinise the functional specificity of recurrent processing for somatosensory awareness.

Another open question concerns the relationship between target detection and subjective experience. Clearly, subjective experience of a detected target at near-threshold intensities is not identical to that of a stimulus at much higher intensities and thus, it is not well modelled by the detection model. Likewise, subjective experience can vary for stimuli at identical physical stimulus intensities and thus, it is not well modelled by the intensity or detection probability models. To address the subjective dimensions of our task, we would require trial-by-trial awareness ratings (and likewise, confidence ratings to evaluate metacognitive aspects of uncertainty). In the current investigation, we did not acquire such ratings because, although they would endow us with a better model of subjective experience, they would also reintroduce the requirement for explicit reports on the stimuli and result in higher relevance of detected compared to undetected targets. However, building on the results of the current study and the opportunities offered by the Bayesian analysis approach, incorporating such subjective reports and probing underlying neural responses may be a promising avenue for future research. In fact, it has been shown that subjective somatosensory awareness is parametrically encoded in connectivity patterns between somatosensory and higher order regions (*Auksztulewicz and Blankenburg, 2013*). This line of research could potentially consolidate experimental findings arguing for local versus global perspectives.

In summary, our findings dissect the functional contribution of different regions in the target detection network and advocate more complex experimental paradigms to dissociate neural responses reflecting conscious access from those supporting collateral functions. Note, however, that this is not to say that frontal and posterior parietal areas do not contribute to the full phenomenal experience of the task. We should keep in mind that although the somatosensory detection task allows for an operationalisation of basic stimulus awareness, detection of simple electrical targets is a simplistic reduction of what constitutes the richness of our everyday conscious perception.

Certainly, cognitive processes such as attention and introspection considerably influence our experience and may even alter perceptual contents and likewise, detection of more complex stimuli, such as tactile motion or objects may require additional cognitive resources further downstream the somatosensory hierarchy. Having said that, when it comes to the perceptual integration of somatosensory stimuli as simple as electrical pulses to the wrist, our data suggest that activity in secondary somatosensory cortex is the best correlate of perceptual success.

## Materials and methods

### Participants

Thirty-two healthy, right-handed volunteers with normal or corrected-to-normal vision completed the experiment. Data of five participants were excluded from the sample because they did not show stable psychometric functions (for more details see Behavioural data analysis and *Figure 2*), leaving data of 27 participants that entered the analyses (18 females, nine males, age range: 19–38). All participants gave written informed consent prior to the experiment and received a monetary reimbursement for their participation. The study was approved by the local ethics committee at the Freie Universität Berlin and complied with the Human Subjects Guidelines of the Declaration of Helsinki.

### Procedure

All participants completed a 30-min training to familiarise with the electrical stimulation and ensure full understanding of the task. In the beginning of the fMRI scanning session, individual psychometric functions were determined to obtain appropriate stimulus intensities for the main task (see Stimulus intensities). Following this ~9 min procedure, all participants completed four runs of the target detection task, each lasting 12.6 min. On each run, 100 experimental trials were presented in random order, interspersed with 10 null events, in which participants fixated throughout the trial without visual or electrical stimulation. The number of trials presented at each intensity level followed a normal distribution to maximise the number of trials with intensities close to detection threshold. This procedure resulted in a total of 400 experimental trials per participant with 64 trials each for the threshold intensity levels 5 and 6, and 16 trials each for the lowest and highest intensity levels 1 and 10.

### Trial design

Each trial was preceded by a variable fixation period of 2–7.5 s, during which participants fixated on a central fixation point, surrounded by a grey disk (*Figure 1A*). The trial started with the presentation of an electrical pulse at one of ten predetermined stimulus intensity levels, which was either detected or missed by the participant. Simultaneously, the grey fixation disk changed its luminance to either white or dark grey, serving as the visual matching cue. A white disk signalled stimulus presence, a dark disk stimulus absence. The two luminance levels of the matching cues as well as the intermediate luminance level presented during fixation were clearly discernible and lasted for 0.8 s. Participants compared their somatosensory percepts (target detected or missed) to the visual matching cues (signalling stimulus presence or absence) and decided on a match or mismatch between the two modalities (*Figure 1A*, inset). After a delay of 0.3 s, participants reported their decision by making a saccade to one of two peripherally presented, colour-coded disks, representing a match or mismatch, respectively. If participants failed to give their responses within 0.9 s, the fixation disk briefly turned red signalling a missed trial.

Importantly, the matching cues were counterbalanced over intensity levels and randomised across trials, resulting in about 50% match and 50% mismatch reports for each intensity level. As a result, any associations of overt reports with stimulus intensity or stimulus detection were eliminated. Likewise, the specific sides on which the colour-coded response cues were presented alternated over trials and were counterbalanced over intensity levels and matching cues to preclude systematic lateralisation of the motor responses. The specific colour code (i.e. the mapping between pink/cyan colours and match/mismatch reports) was counterbalanced across participants.

## Stimulus intensities

Stimulus intensities for the target detection task were drawn from individual psychometric functions that were determined prior to the main experiment. While lying in the MRI scanner, participants were presented with 15 stimulus intensities differing by increments of 0.1 mA and centred on initial estimates of individual 50% detection thresholds (obtained by testing several intensities manually). Each intensity level was repeated 20 times resulting in a total of 300 pulses that were presented in random order. On each trial, participants indicated if they had felt the stimulus or not by pressing one of two buttons. A logistic function with two parameters (50% detection threshold and slope at detection threshold) was fitted to the data yielding a continuous model of the individual psychometric function (*Wichmann and Hill, 2001*). This model was used to obtain estimates of the stimulus intensities resulting in 1% detection probability (T01), 50% detection probability (the individual detection threshold, T50), and 99% detection probability (T99). The 10 stimulus intensities used in the main experiment were spaced equidistantly around T01 (set as intensity level 2) and T99 (set as intensity level 9). This procedure accommodates individual variation in the shape of psychometric functions and was used to ensure a complete sampling of each participant's dynamic range even in case of small drifts in detection thresholds. On average, the procedure yielded initial thresholds of T01 = 1.84 ± .53 mA, T50 = 2.40 ± .69 mA, and T99 = 2.96 ± .92 mA (mean ± standard deviation).

## Stimuli and materials

Electrical stimuli were generated as analogue voltage signals using a waveform generator (DT-9812, Data Translation, Bietigheim-Bissingen, Germany) controlled with Matlab (The MathWorks, Inc, Natick, MA, RRID:SCR_001622). A bipolar constant current stimulator (DS5, Digitimer, Hertfordshire, UK) converted the voltage signal into direct current monophasic square wave pulses of 200 μs duration and administered the stimuli to the left median nerve via MR-compatible adhesive electrodes (GVB-geliMED GmbH, Bad Segeberg, Germany). Responses were recorded using an MR-compatible eye tracker (EyeLink 1000, SR Research Ltd, Mississauga, Ontario, Canada, RRID:SCR_009602). Gaze direction was evaluated online and a response was registered as soon as the gaze remained within the response area for 200 ms. Stimulation and response collection were implemented in Matlab using the Psychophysics (*Brainard, 1997*, RRID:SCR_002881) and EyeLink (*Cornelissen et al., 2002*) toolboxes.

## fMRI scanning procedure

All participants were scanned at the Center for Cognitive Neuroscience Berlin using a 3T Siemens Tim Trio MRI scanner equipped with a 32-channel head coil. T2*-weighted images were acquired using an echo-planar imaging (EPI) sequence (TR = 2000 ms, TE = 30 ms, voxel size = $3{\times}3{\times}3$ mm3, matrix = $64{\times}64$, 37 slices, 20% gap, flip angle = 70°). 378 volumes were obtained on each experimental run. T1-weighted structural images were acquired for coregistration using a 3D MPRAGE sequence (TR = 1900 ms, TE = 2.52 ms, voxel size = $1{\times}1{\times}1$ mm3, FOV = $256{\times}256$ mm2, 176 slices, flip angle = 9°). Including preparation time, estimation of the psychometric function, four experimental runs, and the structural scan, scanning time summed up to approximately 1.5 hr.

## Data analysis

### Behavioural

Behavioural data analysis was performed with Matlab. Models of individual psychometric functions were obtained by fitting logistic functions to the behavioural data of all runs. An average psychometric function was determined for each participant by taking the mean of the fitted detection thresholds and slopes for individual runs. From these average psychometric functions, each participant's fitted detection probabilities for intensity levels 1 and 10 were determined. Five participants, whose detection probabilities were >10% for intensity level 1or <90% for intensity level 10, were excluded from further analysis because for these participants, sampling of the full dynamic range of their psychometric functions was not successful (potentially due to pronounced drifts in detection thresholds, changes in response criteria, or inaccurate reports) (*Figure 2*). Differences in reaction times between detected and undetected targets were assessed using a Bayesian equivalent of the paired t-test (*Krekelberg, 2019*), and the Bayes factor quantifying the evidence for a mean deviation from zero (BF10) is reported. To test if target detection and match/mismatch reports were indeed

independent, we calculated Bayesian tests of association (*Johnson and Albert, 2006*) between these two variables for every participant and report Bayes factors for association (BF10) and independence (BF01). Following the guidelines by *Kass and Raftery (1995)* we consider 1 < BF < 3 negligible, 3 < BF < 20 positive, 20 < BF < 150 strong, and 150 < BF very strong evidence.

## FMRI

FMRI preprocessing and data analysis were performed with SPM12 (www.fil.ion.ucl.ac.uk/spm, RRID: SCR_007037) and custom Matlab scripts. Functional images were realigned using six parameter rigid body transformation to account for head motion, corrected for differences in slice acquisition time, and normalised to standard MNI space using SPM's unified segmentation. Structural images were coregistered to the mean functional image and white matter (WM) and cerebrospinal fluid (CSF) masks were stored for later analysis. Spatial smoothing was not performed prior to the first-level analysis because the Bayesian GLM approach estimates the smoothness of experimental effects from the data using spatial priors (*Penny et al., 2005*) (for information on smoothing of evidence maps and beta images, see below).

Five trial-wise experimental regressors were extracted from the paradigm: 1. The linear stimulus intensity modelling physical stimulus properties, 2. binary target detection as inferred from participants' reports and the presented matching cues modelling an all-or-nothing response to detected targets, 3. detection probability as modelled by individual psychometric functions, 4. expected stimulus uncertainty modelled as the slope of individual psychometric functions (an inverse u-shaped function), and 5. binary match/mismatch responses as a model of overt reports (*Figure 1B*). Five different GLMs were then constructed, each incorporating one of the five experimental regressors: each GLM contained one onset regressor modelling all trial onsets. This onset regressor was parametrically modulated by a z-scored experimental regressor. Z-scored reaction times were added as a further parametric regressor to ensure that differences in model fit could not result from variations in reaction times. Temporal derivatives, motion parameters, as well as the first five principal components explaining variance in the white matter and cerebrospinal fluid signals, respectively were added as nuisance regressors. All models were fitted to each participant's fMRI data using the first-level Bayesian estimation procedure as implemented in SPM12. As spatial prior, we used the recommended Unweighted Graph Laplacian prior, which softly constrains effects to be similar in neighbouring voxels (where the strength of the constraint for each regressor is estimated from the data). With this procedure, we obtained posterior probability maps (*Penny et al., 2005*) for every participant and model, along with free energy approximations to the model evidence in the form of whole-brain voxel-wise log evidence maps (*Penny et al., 2007*). These evidence maps were smoothed with an 8 mm FWHM Gaussian filter, resampled to $2\times2\times2$ mm$^3$ voxel size, and subjected to voxel-wise random effects BMS resulting in one EP map per model for group-level inference (*Rosa et al., 2010*; *Stephan et al., 2009*). Voxels showing EP $\geq$.99 for any model were considered voxels of interest. Since the models were identical except for their experimental regressors, differences in model evidence could only arise from differences in the experimental regressors. Note that we did not include an explicit null model because voxels that were not well explained by any of the models would simply not yield high EPs.

To prevent model dilution for models explaining shared variance, we combined the intensity, detection, and detection probability models into one model family, which adjusts their prior expectation. We call this family the +family because their respective experimental regressors correlate positively with stimulus intensity. We then performed family-level BMS (*Penny et al., 2010*) on the +family, the uncertainty model, and the report model using the .99 EP threshold. Voxels in which the +family explained the data with high probability were defined as ROIs and stored as a +family mask. We then reran the BMS procedure on voxels within the +family mask, this time only comparing the intensity, detection, and detection probability models to determine their individual contributions to the model family fit. To assess the impact of the employed spatial prior and smoothing parameters on the overlap of effects, we repeated the analyses, once using a global shrinkage prior (which does not constrain the smoothness of the data) in the GLM estimation, and once using a 4 mm FWHM smoothing kernel on the evidence maps used for BMS. Neither of these methods considerably reduced the overlap of effects and we report the results using the original parameters. Since we were interested in the behaviour of well-defined regions, we only considered clusters of k $\geq$50 voxels as regions of interest in all analyses.

Next, we tested the beta estimates of the winning models' experimental regressors for systematic deviation from zero. For this purpose, we obtained one group mask for each ROI from the group level EP maps. For the +family models, these masks included all voxels within the .99 EP family clusters in which the respective model scored higher EPs than the other +family models. For the uncertainty and report ROIs, masks were extracted at the .99 EP level. Within each group mask, we determined the subject-level probability peak of the model defining that ROI for every participant (calculated as the ratio of model evidence for that model over the summed model evidence across all models). We then extracted individual beta estimates from these peaks using the beta images of the respective winning models that were previously obtained from the first-level Bayesian GLM estimation and tested their deviation from zero using the Bayesian equivalent of a one-sample t-test (*Krekelberg, 2019*).

To extract SRPs, we defined a new GLM with ten onset regressors, one for each intensity level. Again, the model contained reaction times and all nuisance regressors from the main analysis and was fit to the fMRI data using the first-level Bayesian estimation scheme. The resulting beta images were smoothed at 8 mm FWHM and resampled to $2{\times}2{\times}2$ mm$^3$. For each ROI (as defined by the main analysis), beta estimates for the ten intensity regressors were then extracted from 4 mm radius spheres, centred on the previously identified individual model probability peaks. For each intensity level, the mean beta estimate across the sphere was saved for every participant and the mean beta estimates across participants were plotted as a function of intensity level, yielding SRPs for each region. This way of defining individual spheres ensures their centres lie within the ROI while accommodating individual variation in exact peak locations.

Finally, we examined the distribution of voxels best explained by the different models of the +family across the known cytoarchitectonic subregions of primary and secondary somatosensory cortex (*Eickhoff et al., 2006b*; *Geyer et al., 1999*; *Grefkes et al., 2001*). To this end, we determined which of the +family models yielded the highest exceedance probability in the group level BMS for each voxel within the identified SI and SII ROIs. For each cytoarchitectonic subregion (BA 3b, 1, 2, OP1-4), we then determined the proportion of voxels labelled by the respective +family models to obtain a descriptive summary of the BMS results across regions.

All anatomical coordinates are provided in MNI space. We used the SPM Anatomy Toolbox (*Eickhoff et al., 2005*, RRID:SCR_013273) for anatomical reference where possible and MRIcron (www.mccauslandcenter.sc.edu/mricro/mricron, RRID:SCR_002403) to display fMRI results.

## Acknowledgements

We thank Till Nierhaus for excellent technical assistance.

## Additional information

### Funding

| Funder | Grant reference number | Author |
| --- | --- | --- |
| Deutsche Forschungsgemeinschaft | GRK 1589/2 | Pia Schröder |

The funders had no role in study design, data collection and interpretation, or the decision to submit the work for publication.

### Author contributions

Pia Schröder, Conceptualization, Data curation, Software, Formal analysis, Funding acquisition, Validation, Investigation, Visualization, Methodology, Writing—original draft, Project administration, Writing—review and editing, Interpretation of results; Timo Torsten Schmidt, Conceptualization, Investigation, Writing—review and editing, Interpretation of results; Felix Blankenburg, Conceptualization, Resources, Supervision, Funding acquisition, Project administration, Writing—review and editing, Interpretation of results

## Author ORCIDs

Pia Schröder (iD) http://orcid.org/0000-0002-6135-0691
Timo Torsten Schmidt (iD) http://orcid.org/0000-0003-1612-1301

## Ethics

Human subjects: The study was approved by the local ethics committee at the Freie Universität Berlin (internal reference number: 51/2013) and all participants gave written informed consent prior to the experiment.

## Decision letter and Author response

Decision letter https://doi.org/10.7554/eLife.43410.019
Author response https://doi.org/10.7554/eLife.43410.020

## Additional files

### Supplementary files

• Transparent reporting form
DOI: https://doi.org/10.7554/eLife.43410.013

### Data availability

In accordance with EU's General Data Protection Regulation we are unable to share raw fMRI data. However, single subject log evidence maps and group posterior probability maps have been uploaded to figshare (https://doi.org/10.6084/m9.figshare.7347167.v1) and posterior probability maps can be directly inspected on Neurovault (https://neurovault.org/collections/4496/). Analysis code is available on GitHub (https://github.com/PiaSchroeder/SomatosensoryTargetDetection_fMRI; copy archived at https://github.com/elifesciences-publications/SomatosensoryTargetDetection_fMRI).

The following datasets were generated:

| Author(s) | Year | Dataset title | Dataset URL | Database and Identifier |
|---|---|---|---|---|
| Schröder P | 2018 | Somatosensory target detection | https://neurovault.org/collections/4496/ | Neurovault, 4496 |
| Schröder P | 2018 | Neural basis of somatosensory target detection | https://doi.org/10.6084/m9.figshare.7347167.v1 | Figshare, 10.6084/m9.figshare.7347167.v1 |

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
