## [Decision Letter]

Thank you for submitting your article "Neural basis of somatosensory target detection in humans." for consideration by *eLife*. Your article has been reviewed by two peer reviewers, including Floris de Lange as the Reviewing Editor and Reviewer #1, and the evaluation has been overseen by Joshua Gold as the Senior Editor. The following individual involved in review of your submission has agreed to reveal their identity: Tristan A Bekinschtein (Reviewer #2).

The reviewers have discussed the reviews with one another and the Reviewing Editor has drafted this decision to help you prepare a revised submission.

Summary:

The authors examine the neural correlates of somatosensory target detection, by applying median nerve stimulation electrical pulses of varying strength to human volunteers (N=27). They go beyond earlier investigations of sensory awareness by controlling for several factors that are associated with sensory awareness, i.e. stimulus uncertainty, behavioral relevance, overt reporting and motor responding. They observe that neural responses that can be specifically linked to target detection reside in secondary somatosensory cortex (SII), arguing for a specific role of sensory systems in generating awareness (instead of a broadcasting signal in fronto-parietal cortex). This is a well motivated, well executed and well written piece on a topic of general interest. It is notoriously difficult to disentangle awareness-related activity modulations from its precursors and consequences. I believe the authors have brought this quest a step further, by controlling for several relevant 'confounding factors' and using Bayesian Model Selection in an attempt to disentangle the respective contributions of several factors to the activity profile of sensory and non-sensory regions.

Essential revisions:

1) Anatomical localization of effects

The neural locus of intensity, P(detection) and detection effects are sometimes incredibly close to each other. There is no mention of the smoothing kernel of the fMRI data (please add!), but this does raise the question of what causes the regions of shared variance (intermixed colors). Are they really sensitive to all these effects, or is the intermixing the result of spatial and/or normalization-induced smoothing?

2) Smooth transformation from sensation to perception

The authors argue that there is a smooth transformation from physical to perceptual representation in the somatosensory system. It would be nice if this could be better visualized, and more formally tested – for example by testing whether peak anatomical coordinates for the different task components have a monotonic progression on the posterior-anterior axis.

3) The title seems a bit general, it read like a review title, please consider some version more accurate that maybe reflects this experiment analysis, and conclusions?

4) Several possible confounding factors were raised that should be at least discussed further, or possibly included in new analyses:

a) Perception of near threshold stimuli is difficult, and resolution of associated uncertainty and introspective processes may differ between detected and undetected targets (de Lafuente and Romo, 2011). How might this affect the interpretation of your results?

b) Target detection is the explicit behavioural goal of the task and therefore, detected targets have higher behavioural relevance than undetected targets (Farooqui and Manly, 2018). How might this affect the interpretation of your results?

c) Target detection is directly mapped to overt reports that allow for assessment of participants' trial-by-trial perception (Tsuchiya, Wilke, Frässle, and Lamme, 2015). Can this be teased apart from a possibly covert perceptual process?

d) Overt reports are often communicated with button presses by one hand while stimulation occurs on the other hand, which may affect cortical excitability in homologue regions of the sensorimotor homunculus (Zagha, Casale, Sachdev, McGinley, and McCormick, 2013).

5) Subsection “Behaviour”, the last phrase "Chi squared tests of independence showed no association of target detection with overt reports for any of the participants (all p > .2), confirming that our task design rendered these variables independent."

In the manner I interpret it seems that the authors are assuming that a lack of significance in the test is evidence for a no association, please either rephrase to say the test did not show enough evidence for an association or that there were not significantly associated; alternatively, run an appropriate test pointing to evidence for the null effect (enough evidence of no association) like a bayes factor.

6) In the subsection “fMRI”, I am happy to see that the activity after the BMS is not enough for the authors to claim neural association, they are also demanding fit to the each model (dichotomous, sigmoideal). In this respect they say: "The lack of such a relationship may suggest that the detection-related response that drives the model fit in these regions does not constitute an all-or-nothing response to target detection but may be restricted to only a subset of trials (see Discussion)". I thought that maybe it was also restricted to a subset of participants. I was thinking in asking if they author could provide of an impression of each subject fit and variability in the fMRI data. I am curious to understand the contribution of the participants to these interesting results and one way to do it is to plot single subject parameters. Would the authors be happy to think of a manner to show the direction and strength of effect per participant for the main results? I understand this may seem like a lot of work, it shouldn't. The fitting parameters per participant are easy to extract and the so are the bold estimates. This would allow for a good understanding of the variability of the 27 and also for the contributions each make.

Uncertainty here is not formal report uncertainty (subjective measure of uncertainty) but a toy model of expected uncertainty, it would be good for the authors to comment and give some details on this. Which concept of uncertainty do they refer to, some closer to the objective aspect of detection or closer to a metacognitive evaluation? I know it is defined later in the Discussion but a short account early on would help. Thanks.

Figure 5 legend, at the end the authors refer again to a lack of effect as "confirming that overt reports were independent of stimulus intensity and target detection." I actually like the result but I am also aware of the language vs. the test. Does the test perform confirms or give evidence for a no effect? It is truly independence or it is a lack of dependence? Apologies if I start to sound like the stats pedant.

7) On the topic of "Considering that interoceptive signals, such as the heartbeat, are often faint signals, it would be interesting to see how explicit control of perceptual uncertainty might affect these interpretations." Some year ago we also investigated the role of the insula in faint signals (heartbeat, Canales-Johnson et al., 2015), and found that both in SI and SII but more importantly in insula, the performance modulated the signal, but not the -subjective- impression of improvement. This is not to ask the authors to cite this piece, it is only to show the complexity of the interoceptive argument. There are many papers tapping into the heartbeat as a faint sensory signal, this is mainly because it might be that it is in fact not interoception but a secondary feeling on the heart pounding in the chest or the blood in the neck, ears, etc.

8) Discussion, seventh paragraph, maybe it does not capture any frontal or parietal cortex because the stimuli is so simple? I know this is borderline on strawman but it might be that the simple zap on the skin creates very little demand for detection and hence we are under-powered to capture the extent of the network involve as the demands are too little? This is just food for thought, not a really strong view that I hold. It is always difficult with these faint stimuli, and with detection task in general. I have not forgotten that the experimental design here is elegant and partially controls for this possible critique… It is in fact commented by the end of the Discussion.

---

## [Author Response]

Essential revisions:1) Anatomical localization of effectsThe neural locus of intensity, P(detection) and detection effects are sometimes incredibly close to each other. There is no mention of the smoothing kernel of the fMRI data (please add!), but this does raise the question of what causes the regions of shared variance (intermixed colors). Are they really sensitive to all these effects, or is the intermixing the result of spatial and/or normalization-induced smoothing?

When using Bayesian Model Selection, spatial smoothing is usually done after the 1^st^ level analysis. This is because the Bayesian GLM estimation capitalises on spatial priors, which allow estimating the smoothness of experimental effects from the data. Spatial smoothing is then performed on the resulting evidence maps to facilitate group analyses. For this reason, we did not describe the smoothing in the general preprocessing section, but following the description of the first level analysis:

“These evidence maps were smoothed with an 8mm FWHM Gaussian filter, resampled to 2x2x2 mm^3^ voxel size, and subjected to voxel-wise random effects BMS resulting in one EP map per model for group-level inference (Rosa, Bestmann, Harrison, and Penny, 2010; Stephan, Penny, Daunizeau, Moran, and Friston, 2009).”

To increase visibility of this information, we have added a reference to it in the preprocessing section, where readers would likely look for it:

“Spatial smoothing was not performed prior to the 1^st^ level analysis because the Bayesian GLM approach estimates the smoothness of experimental effects from the data using spatial priors (Penny, Trujillo-Barreto, and Friston, 2005) (for information on smoothing of evidence maps and beta images, see below).”

In addition, we have added information on the spatial prior used in the GLM estimation, as well as information on smoothing of beta images for the extraction of SRPs, which was missing beforehand:

“As spatial prior, we used the recommended Unweighted Graph Laplacian prior, which softly constrains effects to be similar in neighbouring voxels (where the strength of the constraint for each regressor is estimated from the data).”

“The resulting beta images were smoothed at 8mm FWHM and resampled to 2x2x2 mm^3^.”

The question whether regions of shared variance are truly responsive to more than one behavioural dimension is very interesting but also notoriously difficult to answer using fMRI. However, to address your question at least to the level of fMRI voxels (3x3x3 mm^3^), we repeated our analysis, once with a global shrinkage prior instead of the spatial UGL prior in the first level GLM estimation and once with a smaller smoothing kernel (4mm FWHM) on the evidence maps for second level BMS. Neither of these approaches produced considerably more distinct results (see below). In fact, using a global shrinkage prior (which does not impose any prior assumption on the smoothness of the data) resulted in more overlap, suggesting that the use of a spatial prior increased the sensitivity of our analysis. Further, we inspected BMS maps of individual participants and found that these were no less smooth than the group results. Following these observations, we carefully conclude that the regions of shared variance are genuine at least to the resolution allowed by conventional group-level fMRI analyses. However, we would like to stress that the many levels of smoothing that are inherent in fMRI analyses (spatial specificity of the BOLD response, averaging over neurons contained within a voxel, and explicit smoothing during data analysis) limit our ability to draw final conclusions regarding the observed overlap of effects.

We have added a brief description of the alternative smoothing approaches to the Materials and methods section of the manuscript.

“To assess the impact of the employed spatial prior and smoothing parameters on the overlap of effects, we repeated the analyses, once using a global shrinkage prior (which does not constrain the smoothness of the data) in the GLM estimation, and once using a 4mm FWHM smoothing kernel on the evidence maps used for BMS. Neither of these methods considerably reduced the overlap of effects and we report the results using the original parameters.”

2) Smooth transformation from sensation to perceptionThe authors argue that there is a smooth transformation from physical to perceptual representation in the somatosensory system. It would be nice if this could be better visualized, and more formally tested – for example by testing whether peak anatomical coordinates for the different task components have a monotonic progression on the posterior-anterior axis.

We did not mean to imply a smooth spatial gradient, please excuse this misunderstanding. We merely observe that more posterior cytoarchitectonic subregions of SI contain more voxels sensitive to detection probability compared to more anterior regions in the group level BMS. To illustrate this observation and clarify that we are referring to distinct regions, we added a supplementary figure (Figure 3—figure supplement 1), which displays the relative number of voxels best explained by the different +family models in different subregions of SI and SII as defined by the Anatomy Toolbox. In addition, we briefly discuss why we do not find a similar organisational principle in SII. Note however, that these are purely descriptive results, since the voxel counts were obtained from the second level BMS maps.

Nonetheless, the idea of a smooth gradient is very interesting, and we agree that our BMS results in SI hint to the presence of such a gradient. However, given the considerations about spatial resolution raised in question one, we do not feel confident in claiming such a gradient based on the current data. Future high-resolution fMRI studies or even invasive recordings may give more insights into this question.

”Within SI, the spatial distribution of voxels explained by the intensity and detection probability models, respectively was found to follow the known cytoarchitectonic subdivisions of the anterior parietal cortex (Figure 3—figure supplement 1). In SII on the other hand, voxels with sensitivity to intensity, detection probability, and detection did not show an apparent organisation along the cytoarchitectonic subdivisions of the parietal operculum.”

”The distribution of these effects did not align with the cytoarchitectonic subdivisions of the parietal operculum, whose distinct functional roles remain largely elusive. The less pronounced somatotopy and large interindividual variability in SII (Eickhoff, Amunts, Mohlberg, and Zilles, 2006; Sanchez Panchuelo, Besle, Schluppeck, Humberstone, and Francis, 2018) may have concealed the exact organisation of the observed effects and further studies are needed to ascertain the specific functional topology.”

“Finally, we examined the distribution of voxels best explained by the different models of the +family across the known cytoarchitectonic subregions of primary and secondary somatosensory cortex (Eickhoff, Schleicher, Zilles, and Amunts, 2006; Geyer, Schleicher, and Zilles, 1999; Grefkes, Geyer, Schormann, Roland, and Zilles, 2001). […] For each cytoarchitectonic subregion (BA 3b, 1, 2, OP1-4), we then determined the proportion of voxels labelled by the respective +family models to obtain a descriptive summary of the BMS results across regions.”

3) The title seems a bit general, it read like a review title, please consider some version more accurate that maybe reflects this experiment analysis, and conclusions?

Thank you for pointing this out. We have changed the title to be more representative of the study. The title now reads:

“Neural basis of somatosensory target detection independent of uncertainty, relevance, and reports”

4) Several possible confounding factors were raised that should be at least discussed further, or possibly included in new analyses:a) Perception of near threshold stimuli is difficult, and resolution of associated uncertainty and introspective processes may differ between detected and undetected targets (de Lafuente and Romo, 2011). How might this affect the interpretation of your results?b) Target detection is the explicit behavioural goal of the task and therefore, detected targets have higher behavioural relevance than undetected targets (Farooqui and Manly, 2018). How might this affect the interpretation of your results?c) Target detection is directly mapped to overt reports that allow for assessment of participants' trial-by-trial perception (Tsuchiya, Wilke, Frässle, and Lamme, 2015). Can this be teased apart from a possibly covert perceptual process?

*d) Overt reports are often communicated with button presses by one hand while stimulation occurs on the other hand, which may affect cortical excitability in homologue regions of the sensorimotor homunculus (Zagha, Casale, Sachdev, McGinley,* and *McCormick, 2013).*

We have added a new paragraph in the Discussion that addresses each of the raised points and discusses in how far our task was successful in controlling for the different potential confounds:

“The aim of the current study was to experimentally dissociate target detection from four potentially confounding processes and hence, preclude these processes as root causes of detection related neural activity. […] In conclusion, we consider the detection-related responses observed in bilateral SII to be largely free of the addressed experimental confounds.”

5) Subsection “Behaviour”, the last phrase "Chi squared tests of independence showed no association of target detection with overt reports for any of the participants (all p > .2), confirming that our task design rendered these variables independent."In the manner I interpret it seems that the authors are assuming that a lack of significance in the test is evidence for a no association, please either rephrase to say the test did not show enough evidence for an association or that there were not significantly associated; alternatively, run an appropriate test pointing to evidence for the null effect (enough evidence of no association) like a bayes factor.

We entirely agree with this comment and apologise for misrepresenting the test results. We have now replaced all statistical tests with Bayesian equivalents and report Bayes factors capturing evidence for deviation of mean reaction times from zero and Bayes factors for association and independence for the relationship between overt reports and target detection.

“A Bayesian equivalence of the paired-sample t-test suggests strong evidence for shorter reaction times for detected than undetected targets (detected: 352.11 ± 50.98ms, undetected: 363.88 ± 55.97ms, difference: 11.77 ± 14.84ms, BF10 = 88.01). […] In contrast, we found positive evidence for independence for all participants (BF01 > 4) except one (BF01 = 2.62).”

“Differences in reaction times between detected and undetected targets were assessed using a Bayesian equivalent of the paired t-test (Krekelberg, 2019) and the Bayes factor quantifying the evidence for a mean deviation from zero (BF10) is reported. […] Following the guidelines by Kass and Raftery, 1995, we consider 1 < BF < 3 negligible, 3 < BF < 20 positive, 20 < BF < 150 strong, and 150 < BF very strong evidence.”

6) In the subsection “fMRI”, I am happy to see that the activity after the BMS is not enough for the authors to claim neural association, they are also demanding fit to the each model (dichotomous, sigmoideal). In this respect they say: "The lack of such a relationship may suggest that the detection-related response that drives the model fit in these regions does not constitute an all-or-nothing response to target detection but may be restricted to only a subset of trials (see Discussion)". I thought that maybe it was also restricted to a subset of participants. I was thinking in asking if they author could provide of an impression of each subject fit and variability in the fMRI data. I am curious to understand the contribution of the participants to these interesting results and one way to do it is to plot single subject parameters. Would the authors be happy to think of a manner to show the direction and strength of effect per participant for the main results? I understand this may seem like a lot of work, it shouldn't. The fitting parameters per participant are easy to extract and the so are the bold estimates. This would allow for a good understanding of the variability of the 27 and also for the contributions each make.

Thank you for this excellent suggestion. Following your advice, we inspected the single subject parameter estimates in more detail and noticed that indeed there are differences between participants in the direction of the effects. In particular, in frontal and parietal regions that show good fit of the detection model but no clear reflection of the psychometric function in their SRPs, some participants show positive beta estimates for the experimental regressor whereas others show negative estimates, leading to an average estimate close to zero and accordingly, unsystematic SRPs. This finding has led us to reconsider a peculiarity of the model selection procedure, which is that it does not take the directionality of effects into account (since it is based on model evidence, which can be driven by both positive and negative effects). This means that regions with mean parameter estimates of zero but large variability across participants can be picked up by the BMS procedure. However, since we were interested in systematic covariation between the BOLD signal and our experimental regressors, we decided to introduce a second step of statistical testing, in which we computed the evidence for the hypothesis that beta estimates of the winning models’ experimental regressors systematically deviated from zero. We then only considered those regions as regions of interest that showed positive evidence for a deviation. For the detection model, this procedure excluded a subregion of right SII, as well as left SFG, left IPL, and L V3 from the regions of interest. In addition, left thalamus and right SMarG were excluded from areas showing a report effect. We added a description of the procedure and its results to the manuscript, as well as plots showing individual beta estimates for all winning models to Figures 3, 4, and 5 and the mean extracted beta estimates with corresponding Bayes factors to Table 1.

To ensure that none of the excluded regions show a bimodal distribution of beta estimates, which could point towards subgroups in the data, we used Hartigan’s DIP statistic to test beta estimates for bimodality. None of the regions yielded a significant result, suggesting that the null hypothesis of unimodality could not be rejected at the .05 significance level (Hartigan, 1985). Further, we used a Lilliefors test as implemented in Matlab to test for normality and again, none of the regions gave a significant test result, suggesting that the null hypothesis of normality could not be rejected. We conclude that there were no subgroups present in the data and that regions without systematic beta estimates were picked up by the BMS due to large variance across participants. All test results are listed in the table below, however, we did not include these statistics in the manuscript because we consider the beta plots in Figures 3, 4, and 5 sufficiently informative.

Test results for Hartigan’s DIP statistic, testing for unimodality and the Lilliefors test, testing for normality of the extracted beta estimates. None of the tests exceeded the significance level of p < .05.

p value for H0: unimodalityp value for H0: normality*Intensity*R SIa.24.43R SIIp.63.45R SIIa.14.12L SIIm.44.50*Detection probability*R SIp.85.50R SII.99.50L SII.93.50*Detection*R SIIi.40.24R SIIs.53.50L SIIl.58.50L SFG.69.13L IPL.89.50L V3.81.50*Uncertainty*SMG/ACC.43.09R AIC.13.26L AIC.12.28*Report*L SMA.90.13L Thalamus.83.40R SMarG.69.50

“Having established regions that were well explained by the +family models, we further examined the underlying model parameters. […] Since the model selection procedure does not account for directionality of the underlying effects (i.e. both positive and negative parameter estimates may contribute to the model evidence), we imposed this second constraint on the identified regions to detect systematic variation across participants and thus, systematic association of the behavioural regressors with the observed BOLD activity.”

“Next, we tested the beta estimates of the winning models’ experimental regressors for systematic deviation from zero. […] We then extracted individual beta estimates from these peaks using the beta images of the respective winning models that were previously obtained from the 1^st^ level Bayesian GLM estimation and tested their deviation from zero using the Bayesian equivalent of a one-sample t-test (Krekelberg, 2019).”

Uncertainty here is not formal report uncertainty (subjective measure of uncertainty) but a toy model of expected uncertainty, it would be good for the authors to comment and give some details on this. Which concept of uncertainty do they refer to, some closer to the objective aspect of detection or closer to a metacognitive evaluation? I know it is defined later in the Discussion but a short account early on would help. Thanks.

Thank you for pointing this out. We have added a brief definition to the first description of the task and we now refer to our uncertainty model as *expected uncertainty* throughout the manuscript. Moreover, we state in the Discussion that confidence ratings would be required to address metacognitive aspects of the task.

“Accordingly, stimuli presented near individual 50% detection thresholds were expected to be associated with higher uncertainty (as defined by larger trial-by-trial variability in target detection) than clearly sub- or supraliminal stimuli.”

“To address the subjective dimensions of our task, we would require trial-by-trial awareness ratings (and likewise, confidence ratings to evaluate metacognitive aspects of uncertainty).”

Figure 5 legend, at the end the authors refer again to a lack of effect as "confirming that overt reports were independent of stimulus intensity and target detection." I actually like the result but I am also aware of the language vs. the test. Does the test perform confirms or give evidence for a no effect? It is truly independence or it is a lack of dependence? Apologies if I start to sound like the stats pedant.

Again, we agree with this remark and have removed the respective statement. The paragraph now refers to the previously established behavioural result and is otherwise restricted to the descriptive nature of the SRPs.

“As expected from the lack of association between overt reports and target detection, SRPs in these areas did not reflect a systematic relationship with stimulus intensity (Figure 5B).”

7) On the topic of "Considering that interoceptive signals, such as the heartbeat, are often faint signals, it would be interesting to see how explicit control of perceptual uncertainty might affect these interpretations." Some year ago we also investigated the role of the insula in faint signals (heartbeat, Canales-Johnson et al., 2015), and found that both in SI and SII but more importantly in insula, the performance modulated the signal, but not the -subjective- impression of improvement. This is not to ask the authors to cite this piece, it is only to show the complexity of the interoceptive argument. There are many papers tapping into the heartbeat as a faint sensory signal, this is mainly because it might be that it is in fact not interoception but a secondary feeling on the heart pounding in the chest or the blood in the neck, ears, etc.

Thank you for pointing us to this interesting paper. We did not mean to imply that the link between insula and interoceptive awareness is fully established, please excuse if this was unclear. In fact, we agree with the conclusions drawn in the paper you mentioned that suggest that signals in the insula may reflect processes that are related to interoceptive processing and influence interoceptive sensitivity but do not reflect awareness. Building on our results, we suggest that incorporating perceptual uncertainty as an explicit behavioural dimension in future studies might be a fruitful approach to further understand the exact role of the insula in the interoceptive process, notwithstanding existing evidence. We have rephrased the respective paragraph in the manuscript and hope that this makes it clearer.

“The insula in particular has been assigned a pivotal role in interoceptive inference (Seth, 2013) and awareness (Craig, 2009; Critchley, Wiens, Rotshtein, Öhman, and Dolan, 2004), although recent research suggests that activity in AIC might reflect interoceptive sensitivity and learning rather than subjective experience per se (Canales-Johnson et al., 2015). In light of our results and given the fact that interoceptive signals, such as the heartbeat, are often faint signals, future research may benefit from combining measures of perceptual uncertainty (e.g. Garfinkel, Seth, Barrett, Suzuki, and Critchley, 2015) with neuroimaging techniques to reconcile interoceptive processing with insular response properties.”

8) Discussion, seventh paragraph, maybe it does not capture any frontal or parietal cortex because the stimuli is so simple? I know this is borderline on strawman but it might be that the simple zap on the skin creates very little demand for detection and hence we are under-powered to capture the extent of the network involve as the demands are too little? This is just food for thought, not a really strong view that I hold. It is always difficult with these faint stimuli, and with detection task in general. I have not forgotten that the experimental design here is elegant and partially controls for this possible critique… It is in fact commented by the end of the Discussion.

This is indeed an interesting question. It is well conceivable, that more complex stimuli would require engagement of different regions and more extended networks and we do not wish to claim that SII is the neural substrate of any kind of conscious somatosensory percept. However, the most popular frontoparietal network theories of conscious perception postulate that any conscious percept occurs with a global ignition and “intense activation spreading to the frontoparietal network” (Dehaene, Changeux, Naccache, Sackur, and Sergent, 2006). Even though in our task the stimuli were simple and relatively faint on threshold trials, participants were arguably conscious of them and definitely so on trials with high intensities. If a global ignition of the frontoparietal network did underlie these conscious percepts, we would expect to find corresponding activation in our results, especially considering that frontoparietal activity is ubiquitous in neuroimaging studies and by no means difficult to detect. The fact that we did not find reliable engagement of the frontoparietal network in target detection, at the very least, does not offer supporting evidence for the frontoparietal network as part of “the minimum neural mechanisms jointly sufficient for any one specific conscious experience” (Koch, Massimini, Boly, and Tononi, 2016). Instead, we find SII as the most likely neural correlate of conscious somatosensory perception during target detection. In the manuscript, we have added the possibility of a more extended network for more complex stimuli but believe that a more comprehensive discussion of this question is beyond the scope of the current study.

“Certainly, cognitive processes such as attention and introspection considerably influence our experience and may even alter perceptual contents and likewise, detection of more complex stimuli, such as tactile motion or objects may require additional cognitive resources further downstream the somatosensory hierarchy.”